

# New correction method of scattering coefficient measurements of a three-wavelength nephelometer

Jie Qiu[1], Wangshu Tan[1,2], Gang Zhao[1,3], Yingli Yu[4,1], Chunsheng Zhao[1]

[1]Department of Atmospheric and Oceanic Sciences, School of Physics, Peking University, Beijing 100871, China
[2]School of Optics and Photonics, Beijing Institute of Technology, Beijing 100081, China
[3]State Key Joint Laboratory of Environmental Simulation and Pollution Control, College of Environmental Sciences and Engineering, Peking University, Beijing 100871, China
[4]Economics & Technology Research Institute, China National Petroleum Corporation, Beijing 100724, China

*Correspondence to*: Chunsheng Zhao (zcs@pku.edu.cn)

**Abstract.** The aerosol scattering coefficient is a significant parameter for estimating aerosol direct radiative forcing, which can be measured by nephelometers. Currently, nephelometers have the problem of non-ideal Lambertian light source and angle
truncation. Hence, the observed raw scattering coefficient data need to be corrected. In this study, based on the random forest machine learning model, we have proposed a new method to correct the scattering coefficient measurements of a three-wavelength nephelometer, Aurora 3000, under different relative humidity conditions. The result shows that the empirical corrected values match Mie-calculation values very well at all the three wavelengths and under all the measured relative humidity conditions, with more than 85 % of the corrected values in error by less than 2 %. The correction method is valid to
obtain scattering coefficient with high accuracy and there is no need for additional observation data.

## 1 Introduction

Atmospheric aerosol particles directly impact the earth radiative balance by scattering or absorbing the solar radiation. However, the aerosol direct radiative forcing varies greatly, ranging between -0.77 and 0.23 W/m$^2$ (IPCC, 2013), which poses a great challenge to the accurate quantification of its effects on the earth climate system. Aerosol single scattering albedo and
extinction coefficient are the two most important parameters for estimating aerosol direct radiative forcing, both depending on the aerosol scattering coefficient and absorbing coefficient. Part of the estimation uncertainty comes from the inaccuracy in the measurement of these aerosol properties. Therefore, more precise measurements are needed. In recent years, two commercial integrating nephelometers (Aurora 3000 and TSI 3563) have been developed to measure aerosol scattering coefficients and hemispheric backscattering coefficients at three different wavelengths (450 nm, 525 nm, 635 nm for Aurora
3000 and 450 nm, 550 nm, 700 nm for TSI 3563). The three-wavelength integrating nephelometer is widely employed in field measurements and laboratory studies due to its high accuracy in measuring aerosol scattering coefficients (Anderson et al., 1996). However, it has two primary drawbacks, namely the angle truncation and nonideal Lambertian light source, contributing





to a certain systematic error (Bond et al. 2009). The angle truncation indicates the lack of illumination near 0° and 180° and the nonideal Lambertian light source means that the measured scattered signal is non-sinusoidal. The two drawbacks render

the nephelometer measurement less precise.

In order to correct the measurement errors of nephelometer, Anderson and Ogren (1998) used a single parameter as the scattering correction factor to quantify the nonideal effects. The scattering correction factor is defined as the ratio of Mie-calculated scattering coefficient to that measured by the nephelometer and is closely related to the aerosol size and chemical composition. Müller et al. (2011) demonstrated that several methods have been proposed to derive it. Initially, researchers

simulated the nephelometer measurements based on the Mie model. In more detail, they replaced the ideal sinusoidal function with nephelometer's actual scattering angle sensitivity function to derive the scattering coefficient under the condition of nephelometer light source. The scattering coefficient under the condition of ideal Lambertian light is also obtained by the Mie model, thereby calculating the correction factor. However, this method needs the additional information of particle number size distribution (PNSD), particle shape and refractive index (Quirantes et al., 2008), and it is not convenient to obtain

simultaneous PNSD data because the measurement instrument is expensive and not easy to maintain.

An alternative popular correction mechanism is to constrain the correction factor simply by the wavelength dependence of scattering (scattering Ångström exponent (SAE)). Considering that SAE and scattering correction factor both rely on particle size, Anderson and Ogren (1998) figured out the linear relationship between them for each TSI nephelometer's wavelength. This ingenious method is convenient, because the scattering properties at different wavelengths, or SAE, can be directly

measured by nephelometer itself. However, Bond et al. (2009) found that SAE is affected by both particle size and refractive index, while correction factor is scarcely impacted by refractive index. Therefore, in the cases when the variation of refractive index can be neglected, SAE could relate to correction factor well; while in the cases when refractive index varies significantly, for example, in the presence of water, the relationship between SAE and correction factor is complicated (Anderson et al. 1996). Furthermore, the absorption properties of sampled particles can alter the wavelength dependence of scattering,

contributing to errors of this correction method for absorbing aerosols (Bond et al., 2009). Therefore, it is not an accurate correction method to use the linear relationship between a single parameter SAE and correction factor.

In this study, the measurement limitations of angle truncation and nonideal Lambertian light source are both considered. In view of the disadvantages of these methods mentioned above, we put forward a new correction method of scattering coefficient measurements of a three-wavelength nephelometer, with the use of machine learning model and Aurora 3000

measurement data. A description of methodology under dry conditions and other relative humidity conditions is given in Sect. 2. The accuracy of the two methods is verified in Sect. 3. At last, the conclusions are presented in Sect. 4.



## 2 Data and Method

### 2.1 Site description

Eight field observations (Table 1)  were conducted at different time periods in China, including two observations in Wuqing
(39°38' N, 117°04' E), two in Xianghe (39°76' N, 117°01' E), each observation in Wangdu (38°40' N, 115°08' E), Zhangqiu
(36°71' N, 117°54' E), Beijing(39°59' N, 116°18' E) and Gucheng (38°9' N, 115°44' E). Five sites (Wuqing, Xianghe,
Wangdu, Zhangqiu, Gucheng) are located in suburban areas, representing the characteristics of regional anthropogenic aerosols
on the North China Plain. Measurement in Beijing was conducted in Peking University (downtown Beijing), surrounded by
two heavy traffic roads, and hence it can well represent the typical case of urban pollution. As shown in Fig. 1, the number
size distributions of our datasets cover a wide range of 10-1000 nm, including most continental aerosol types.

**Table 1.** The summary of eight field observations used in this paper.

| Site | (1)Wuqing | (2)Wuqing | (3)Xianghe | (4)Xianghe | (5)Wangdu | (6)Zhangqiu | (7)Beijing | (8)Gucheng |
|---|---|---|---|---|---|---|---|---|
| Date | 07 March- 04 April | 12 July- 14 August | 22 July- 30 August | 09 July- 08 August | 04 June- 14 July | 23 July- 24 August | 25 March- 09 April | 15 October- 25 November |
| Year | 2009 | 2009 | 2012 | 2013 | 2014 | 2017 | 2017 | 2016 |
| PNSD | TDMPS +APS | TDMPS +APS | SMPS +APS | TDMPS +APS | TDMPS +APS | SMPS +APS | SMPS +APS | SMPS +APS |
| BC | MAAP | MAAP | MAAP | MAAP | MAAP | AE33 | AE33 | AE33 |
| f(RH) | / | / | / | / | TSI 3563 | Aurora 3000 | Aurora 3000 | Aurora 3000 |



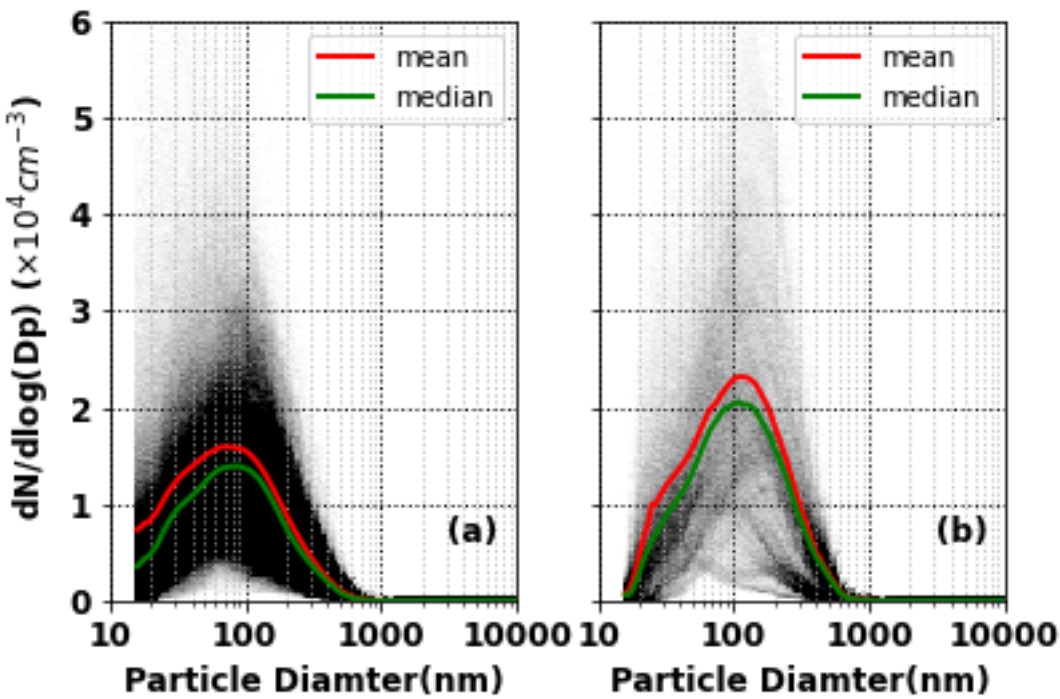

**Figure 1.** The number size distribution of the measured aerosol in the (a) field observation (1)-(7) and (b) field observation (8).

## 2.2 Correction under dry conditions

An important feature of Mie scattering is that the larger the particle, the more forward scattering, meaning that the ratio of backscattering coefficient to total scattering coefficient, or hemispheric backscattering fraction (HBF), would become smaller. Therefore, HBF can to some extent stand for aerosol size. Considering that both SAE and correction factor relate to particle size, this paper uses the datasets of field observation (1)-(7) to explore the relationship between scattering correction factor (hereinafter CF) and calculated SAE and HBF at different wavelengths.

CF could be constrained by SAE to a certain extent, but we cannot fit them with a simple linear regression equation in that it is also related to HBF (Fig. 2). On the whole, the larger the HBF, the greater the slope of CF changing with SAE.



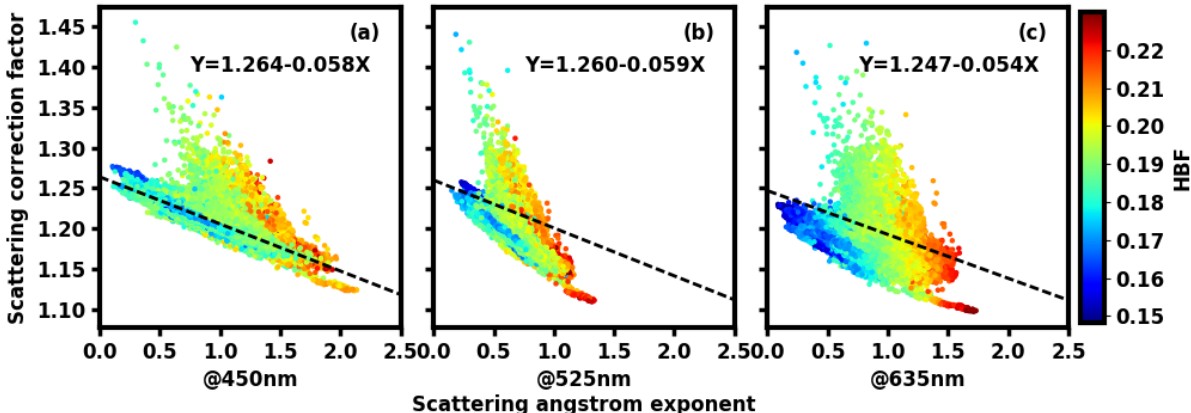

**Figure 2.** Scattering correction factors versus the scattering Ångström index. (a), (b), (c) respectively represent the results at the wavelengths of 450 nm, 525 nm, and 635 nm. The black dashed line is a statistical linear relationship, and the color of points represents the hemispheric backscattering fraction HBF.

Before establishing the relationship between CF and calculated SAE and HBF, it is necessary to figure out the size range represented by SAE and HBF. The paper makes the assumption of three types of particle composition: scattering particles, absorption particles, and core-shell mixed particles with the core radius of 35 nm; based on this assumption and datasets mentioned above, the variation of SAE at the three wavelength combinations (450+525 nm, 450+635 nm, 525+635 nm) and HBF at the three bands (450 nm, 525 nm and 635 nm) in the particle size range (100 nm-10 μm) is calculated by the Mie model. Additionally, to distinguish the particle size range where the change of SAE and HBF can be obviously manifested in the overall optical properties of aerosols, the paper also calculates the ratio of size-resolved scattering and hemispheric backscattering to total scattering for three types of assumed aerosols.

As shown in Fig. 3, for all the three types of aerosols, scattering is mainly concentrated in the size range of 100-1000 nm; while particles larger than 1000 nm contribute little to the total scattering, and hence there is no follow-up discussion of SAE change of these large particles. When particles are smaller than 1000 nm, the overall trend of SAE is decreasing with the increase of particle size. Especially when the particle is greater than 300 nm, with particle size increasing, the decline rate of SAE is relatively large, and SAE calculated at different bands is obviously different. Therefore, the SAE calculated at the three band combinations is approximately representative of particle size ranging from 300 to 1000 nm.

**Figure 3.** The SAE change of scattering particles (a), absorption particles (b), and core-shell mixing particles of core radius 35 nm (c) with the change in particle diameter (solid line). The dashed lines represent the ratio of scattering at a certain diameter relative to the total scattering.





105 From Fig. 4, for environmental aerosol particles, the backscattering of particles in the 100-1000 nm range also contributes a lot to the total scattering, and the HBF characteristics of particles greater than 1000 nm are no longer discussed below. For particles with a size less than 300 nm, all three types of aerosol particles show a noticeable feature of HBF decreasing with the increment of particle size. However, when the particle becomes larger than 300 nm, the HBF is almost unchanged. In other words, HBF can represent the size information of particles smaller than 300 nm.




**Figure 4.** The HBF change of scattering particles (a), absorption particles (b), and core-shell mixing particles of core radius 35 nm (c) with the change in particle diameter (solid line). The dashed lines represent the ratio of hemispheric backscattering at a certain diameter relative to the total scattering.





Based on the above analysis, it is known that SAE and HBF can represent different size information of aerosol particles,
finally deriving the particle size information of 100-1000 nm. Moreover, it should be noted that both SAE and HBF mentioned above are the calculation results from scattering and backscattering coefficients under the Aurora 3000 nephelometer light source condition.

In order to derive accurate scattering and backscattering information which is also affected by the mass concentration of BC and its mixing state with other aerosols, not only PNSD but also black carbon (BC) data are needed to run the Mie model.
According to Ma et al. (2012), when considering the amount of externally mixed BC and core-shell mixed BC, we use $R_{ext}$ to represent the ratio of the mass concentration of the externally mixed BC ($M_{ext-BC}$) to that of the total BC ($M_{BC}$):

$$R_{ext} = M_{ext-BC}/M_{BC}. \tag{1}$$

It is pointed out that $R_{ext}$ is sensitive to HBF. Therefore, on the basis of Mie model, we use PNSD, $M_{BC}$ and the assumed $R_{ext}$ value to calculate HBF. Next, the calculation HBF is compared with the observation result of nephelometers. If their
difference is minimal, the assumed $R_{ext}$ value is considered true. Deriving mass concentration of BC and PNSD data, assuming that the true $R_{ext}$ is consistent at each size and there is no difference in the radius of core-shell mixed particles with the same size, we can calculate the number size distribution of core-shell mixed BC and externally mixed BC. Furthermore, the refractive index can also be obtained, making it possible to derive more precise information of scattering, backscattering and then SAE and HBF. Details about this method of retrieving PNSD and refractive indices can be found in Ma et al.
130    (2012).

Since it is difficult to use a linear regression equation to figure out the relationship between these six physical quantities (three SAE and three HBF) and CF, a random forest machine learning model from the scikit-learn machine learning library (Prettenhofer et al., 2011), an effective method that can be used for classification and nonlinear regression (Breiman, 2001), is adopted. The random forest model has several advantages (Zhao et al., 2018) as follows: First, it involves fewer assumptions
of dependency between observations and results than traditional regression models. Second, there is no need for a strict relationship among variables before implementing model simulation. Third, this model requires much fewer computing resources than deep learning. Finally, it has a lower over-fitting risk.

In brief, our correction method of nephelometers under the dry condition encompasses the following procedures (Fig. 5):

(1) Obtain information on particle number size distribution (PNSD), black carbon (BC), and mixing state ($R_{ext}$) of field
140        observation (1)-(7).

(2) Calculate the scattering and backscattering by Mie model under the conditions of the nephelometer light source at the wavelengths of 450 nm, 525 nm and 635 nm.

(3) Calculate the hemispheric backscattering fraction HBF at the three wavelengths.

(4) Calculate the scattering Ångström index SAE of the three band combinations (450+525 nm, 450+635 nm, 525+635 nm).

(5) Calculate the scattering and backscattering by Mie model under the conditions of the ideal light source at the wavelengths of 450 nm, 525 nm and 635 nm.



(6) Based on the results of the second and fifth steps, calculate theoretical CF at the three wavelengths.

(7) Use six parameters, including three HBF and three SAE, and theoretical CF of each wavelength to train the machine learning model, deriving RF predictor.

(8) Verify the predictive validity of the trained model with the dataset of Gucheng.

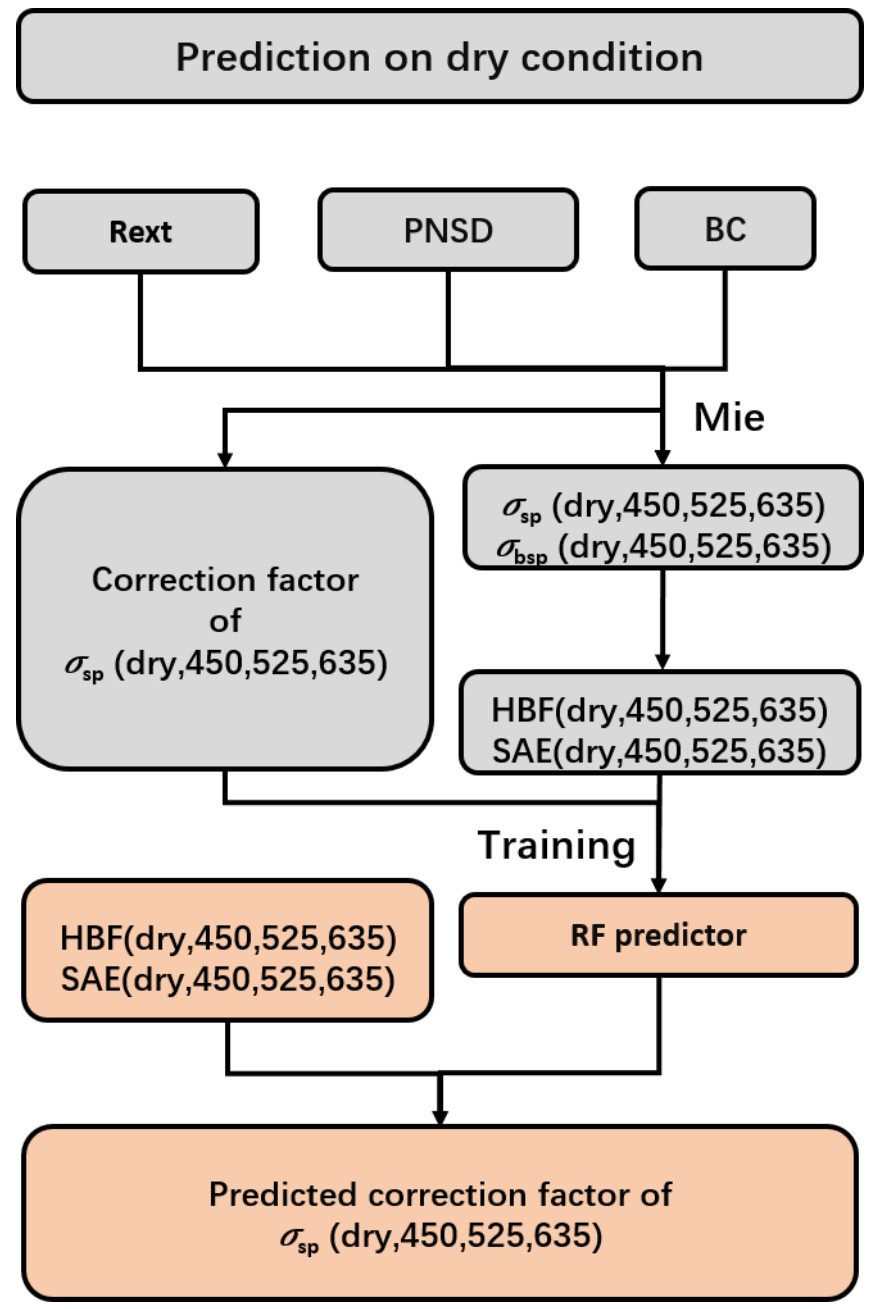

**Figure 5.** Flow chart of estimating CF under dry conditions by machine learning.





**2.3 Correction under different RH conditions**

Müller et al. (2011) established the linear fit relationships between CF and the corresponding SAE at different wavelengths.
By using the data of Gucheng to verify his method, our paper finds that the predicted correction factor gradually becomes
much more different from the theoretical Mie-calculation correction factor as the relative humidity increases, and thereby the
statistical relationship between CF and SAE can no longer represent most cases. Especially under elevated relative humidity
conditions, a correction method taking the hygroscopicity into account is needed, because, with the increment of relative
humidity, the non-absorbing component in the aerosol particle can take up water due to its hygroscopicity and then grow up.
Accordingly, the water content and particle size may change, resulting in a certain change of CF for the same group of aerosol
particles.

The hygroscopicity or aerosol hygroscopic growth could be indicated by the scattering hygroscopic growth curve f(RH)
and the backscattering hygroscopic growth curve $f_b(RH)$: At low relative humidity, the growth due to aerosol taking up water
is weak and thus the change of f(RH) and $f_b(RH)$ is small; as relative humidity goes up, the aerosol hygroscopic growth is
obvious, and particle size changes a lot. Correspondingly, the change of f(RH) and $f_b(RH)$ is large. Referring to researches of
Kuang et al. (2017) and Brock et al. (2016), the following formulas are used to describe f(RH) and $f_b(RH)$:

$$f(RH) = 1 + \kappa_{sca}\frac{RH}{100 - RH}, \tag{2}$$

$$f_b(RH) = 1 + \kappa_{bsca}\frac{RH}{100 - RH}, \tag{3}$$

where $\kappa_{sca}$ and $\kappa_{bsca}$ are fitting parameters representing the hygroscopic growth rate in aerosol scattering and backscattering.
According to the PNSD of outfield observation (1)-(7) and different assumed size distributions of $\kappa$, the theoretical Mie-
calculation values are presented as scatter points in Fig. 6. On the basis of the above formulas, the lines represent fitted curves
under the condition of nephelometer light source. As can be seen, for the three bands, Eq. (2) and Eq. (3) basically describe
the trend of f(RH) and $f_b(RH)$. In other words, aerosol scattering and hemispheric backscattering hygroscopic growth can be
represented by parameters of $\kappa_{sca}$ and $\kappa_{bsca}$. As a result, we wonder whether or not the hygroscopic growth of scattering
correction factor, or $C(RH)$, could be fitted similarly as above formulas with parameter $\kappa_c$. The black scatter points in the
figure do not lie close to the black dashed lines, and accordingly, the fit formula cannot accurately describe $C(RH)$.





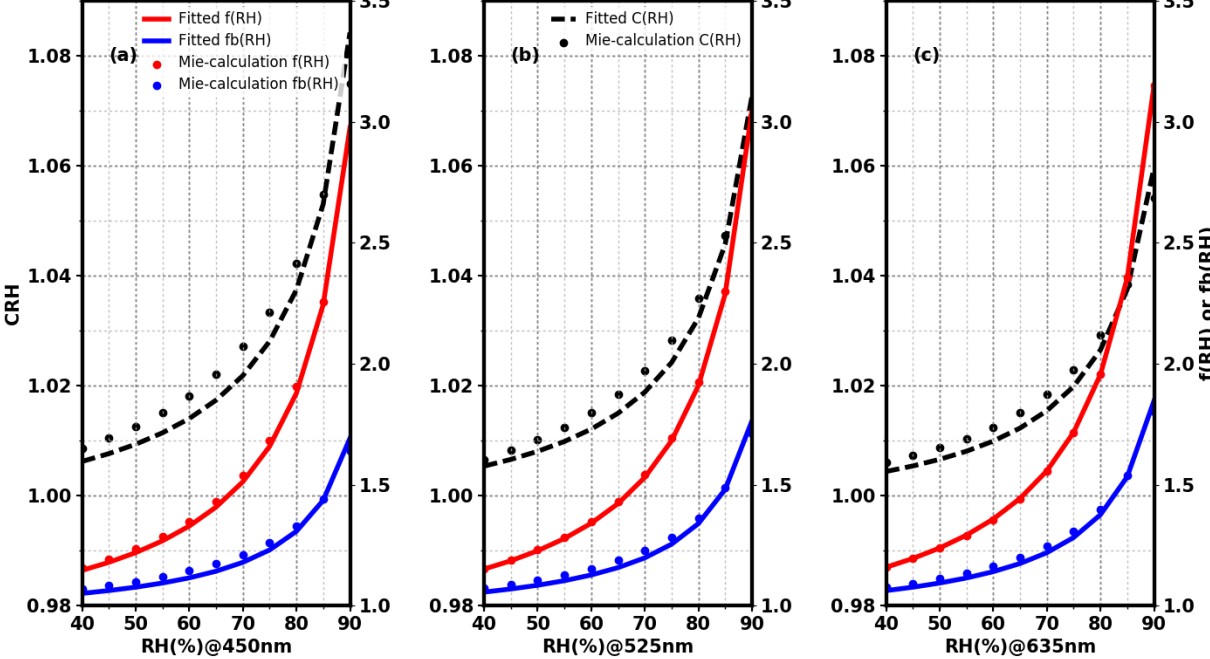

**Figure 6.** The comparison between κ fitted and theoretical Mie-calculation f(RH), $f_b(RH)$ and C(RH) at the wavelengths of 450 nm (a),
525 nm (b), and 635 nm (c) under the condition of nephelometer light source. The scatter points represent each theoretical Mie-calculation
value. The red solid line is the f(RH) fitted curve and the blue solid line is the $f_b(RH)$ fitted curve, both corresponding to the right ordinate
value. The black dashed line is the C(RH) fitted curve, which corresponds to the left ordinate value.

Therefore, this paper attempts to derive CF under different RH conditions in a similar machine learning way as described
for the dry state. First of all, we need to figure out parameters impacting CF under different RH conditions. Aerosol size
accounts for CF, referring to Sect. 2.2, and thereby SAE and HBF in the dry state at three wavelengths are needed. Besides,
hygroscopicity matters to a large extent; κ-Köhler theory (Petters and Kreidenweis, 2007) is thus applied, which uses
hygroscopicity parameter κ to describe the hygroscopic growth of aerosol particles under different relative humidity conditions:

$$S = \frac{D^3 - D_d^3}{D^3 - D_d^3(1-\kappa)} \cdot \exp(\frac{4\sigma_{s/\alpha} \cdot M_{water}}{R \cdot T \cdot D_d \cdot g \cdot \rho_w}). \tag{4}$$

Where S is saturation ratio; D is the diameter of the aerosol particle after hygroscopic growth; $D_d$ is the diameter of the aerosol
particle in the dry state; $\sigma_{s/\alpha}$ is the surface tension at the interface between the solution and air; T represents absolute
temperature; $M_{water}$ is the molar mass of water; R is the universal gas constant and $\rho_w$ is the density of water.

With the information of PNSD, refractive index of dry aerosol, mixing state, size distribution of κ, and water refractive
index of $1.33 - 10^{-7}$i (Seinfeld and Pandis, 2006), on the basis of κ-Köhler theory (Eq. (4)), we can calculate the aerosol
optical parameters at different RH, deriving f(RH) and $f_b(RH)$. Next, Eq. (2) and Eq. (3) are used to fit the curve of f(RH)





and $f_b(RH)$ at each wavelength, deriving fitting parameters $\kappa_{sca}$ and $\kappa_{bsca}$ which can imply the size-resolved hygroscopicity. Combined with relative humidity, the estimated change of CF with the relative humidity involves up to 13 physical quantities.

To summarize, our correction method of nephelometers under different relative humidity conditions encompasses the following procedures (Fig. 7):

(1) Obtain information on particle number size distribution (PNSD), black carbon (BC), mixing state ($R_{ext}$), aerosol hygroscopicity parameter ($\kappa$), and relative humidity RH of field observation (1)-(7).

(2) Calculate the scattering and backscattering by Mie model under the conditions of the nephelometer light source at the wavelengths of 450 nm, 525 nm and 635 nm in the dry state.

(3) Calculate the hemispheric backscattering fraction HBF at the three wavelengths under dry conditions.

(4) Calculate the scattering Ångström index SAE of the three band combinations (450+525 nm, 450+635 nm, 525+635 nm) under dry conditions.

(5) Under different relative humidity conditions and assumptions of aerosol hygroscopicity, according to the $\kappa$-Köhler theory, aerosol scattering and hemispheric backscattering after the hygroscopic growth are calculated on the basis of nephelometer light source at three wavelengths.

(6) Calculate f(RH) and $f_b(RH)$ curves of the three bands based on the scattering and hemispheric backscattering under dry and different relative humidity conditions.

(7) Calculate the fitting parameters of $\kappa_{sca}$ and $\kappa_{bsca}$ from f(RH) and $f_b(RH)$.

(8) Calculate the scattering and hemispheric backscattering after the hygroscopic growth under the conditions of the ideal light source at three wavelengths.

(9) Based on the results of the fifth and eighth steps, calculate theoretical CF at the three wavelengths.

(10) Use thirteen parameters, including three HBF and three SAE, relative humidity RH, three $\kappa_{sca}$ and three $\kappa_{bsca}$ at this RH, and theoretical CF of each wavelength to train the machine learning model, deriving the RF predictor.

(11) Verify the predictive validity of the trained model with the dataset of Gucheng.



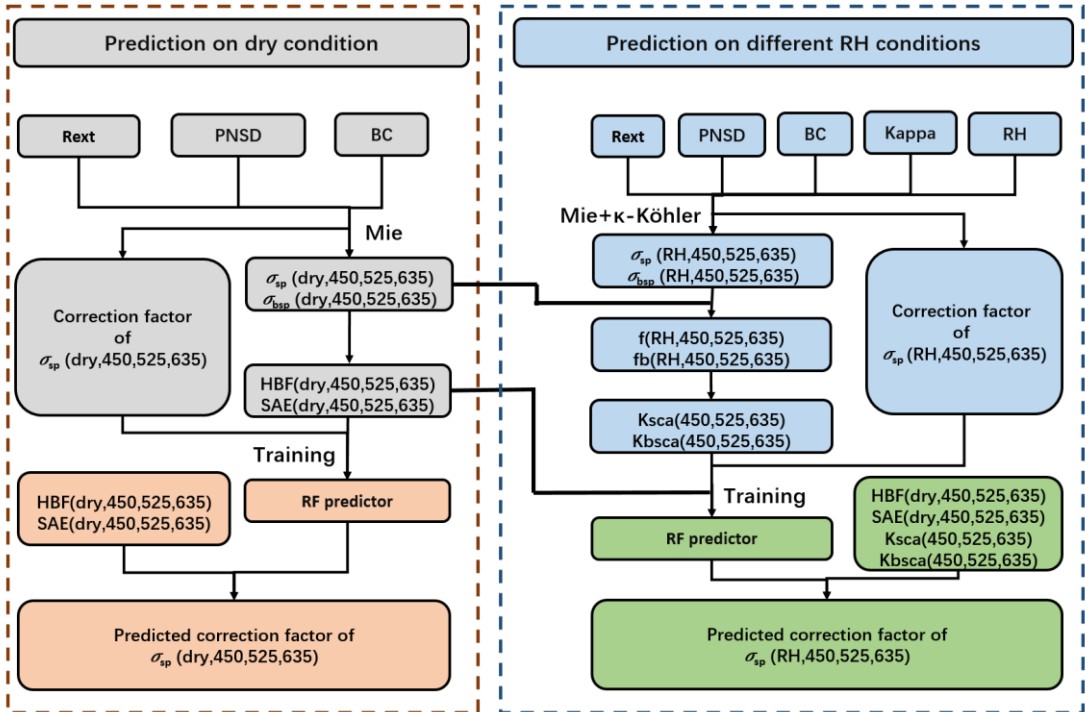

**Figure 7.** Flow chart of estimating CF under different relative humidity conditions by machine learning.

## 3 Results and discussions

In order to verify the methods introduced above, on the basis of Gucheng data and the derived RF predictor, we have predicted CF and compared it with the theoretical Mie-calculated CF.

### 3.1 Under dry conditions

As can be seen from Fig. 8, for 450 nm and 525 nm, the prediction performance is relatively good, and the correlation coefficient between prediction value and the theoretical Mie-calculation is 0.88 and 0.84, respectively; more than 90 % of the points fall within the error range of 2 %, and most of them are basically concentrated near the 1:1 line. For 635 nm, the result is slightly worse, with the correlation coefficient at 0.76 and 85.88 % of points in error by less than 2 %. In general, compared with the traditional correction method, our method does not need to consider whether or not the aerosol has strong or wavelength dependent absorption, which improves the accuracy of the CF estimation in the dry state; in addition, the input parameters can be obtained by the nephelometer's observation.





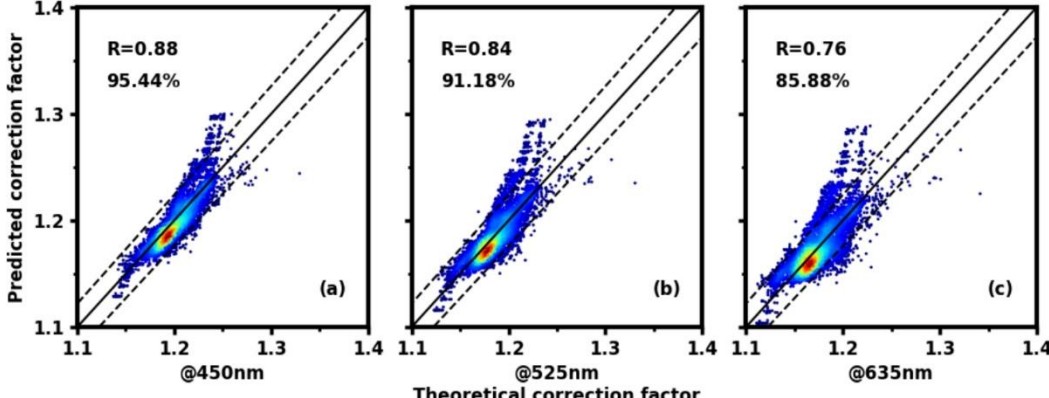

**Figure 8.** On dry conditions, the comparison of the correction factors calculated by our method and theoretical Mie-calculation values at
the wavelengths of 450 nm (a), 525 nm (b), and 635 nm (c), respectively, with a black solid 1:1 line and two dashed lines representing a
deviation of 2 %. The color of the data point represents data density; the warmer the hue, the denser the data point. R is the correlation
coefficient, and the percentage indicates the percentile of points falling within the error range of 2 %.

## 3.2 Under different RH conditions

The paper uses each PNSD of the field observation (1)-(7) and averages them to plot Fig. 9 which represents the variation
characteristics of CF with the change of relative humidity and aerosol population hygroscopicity, at three wavelengths of 450
nm, 525 nm and 635 nm, respectively. When it comes to the aerosol hygroscopicity, according to 24 size distributions of κ
obtained from Hachi field observation (Liu et al., 2014), the paper takes their average size distribution (the total volume-
weighted κ is 0.281) as the basis; next, in order to obtain a sequence of size distributions of κ, the basis κ is multiplied from
0.05 to 2, with 0.01 as the interval. Therefore, different colors in the figure indicate the overall hygroscopicity of different
aerosols.

As shown in Fig. 9, under all the different relative humidity conditions, CF of the 450 nm is the largest, with that of 525
nm coming second, and that of 635 nm is the smallest. All CFs at the three bands increase with the increment of relative
humidity. Furthermore, if the relative humidity remains constant, CF also increases with aerosol hygroscopicity increasing.
Therefore, the results support the view that both the environment relative humidity and the hygroscopicity of aerosols impact
CF.



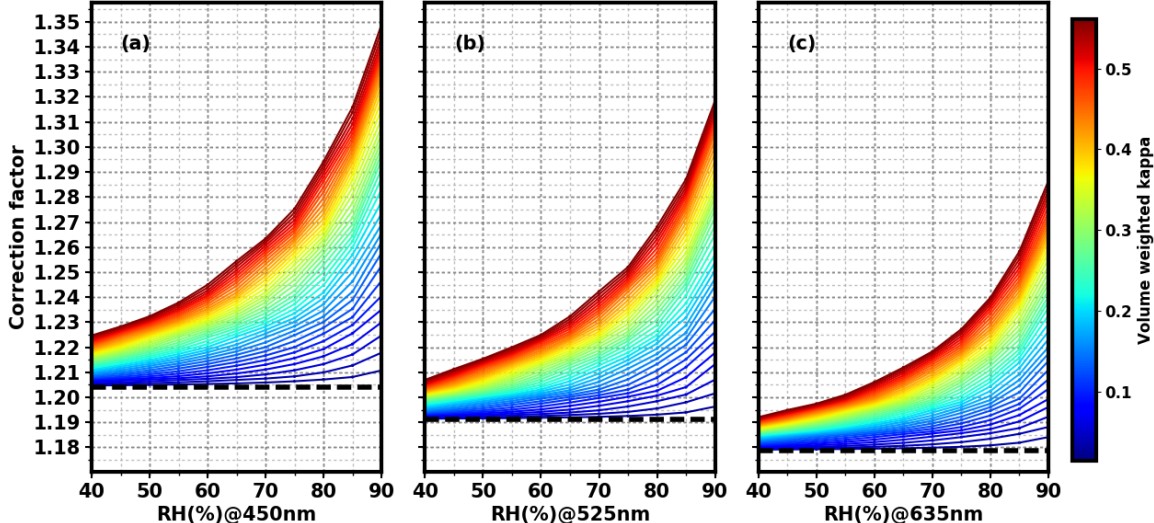

**Figure 9.** The theoretical calculation of the scattering correction factors (CF) versus relative humidity (RH) and hygroscopicity κ at the wavelengths of 450 nm (a), 525 nm (b), 635 nm(c). The dashed line represents scattering correction factor in the dry state, and the color represents the hygroscopicity κ. The color bar is derived from multiplying the total volume-weighted κ of 0.281 by 0.05 to 2, with 0.01 as the interval.

Our correction method under different RH conditions takes the humidity and hygroscopicity into account. As depicted in Fig. 10, the new method predicts CF very well at all the three wavelengths, and nearly all scatter points at the three wavelengths are centered near the 1:1 line. For the 450 nm band, the correlation coefficient between prediction value and theoretical Mie-calculation reaches 0.99, with 99.54 % of the points falling within the error range of 2 %; for the 525 nm band, the correlation coefficient is 0.98, with 98.99 % of the points falling within the error range of 2 %; for the 635 nm band, the correlation coefficient is 0.95, with 96.37 % of the points in error by less than 2 %. From graphs of (d), (e) and (f), the new method's estimation of CF is basically consistent in accuracy at each relative humidity. Another advantage of our new method is that all these input parameters can be obtained by the nephelometer's observation, achieving the goal of self-correction.





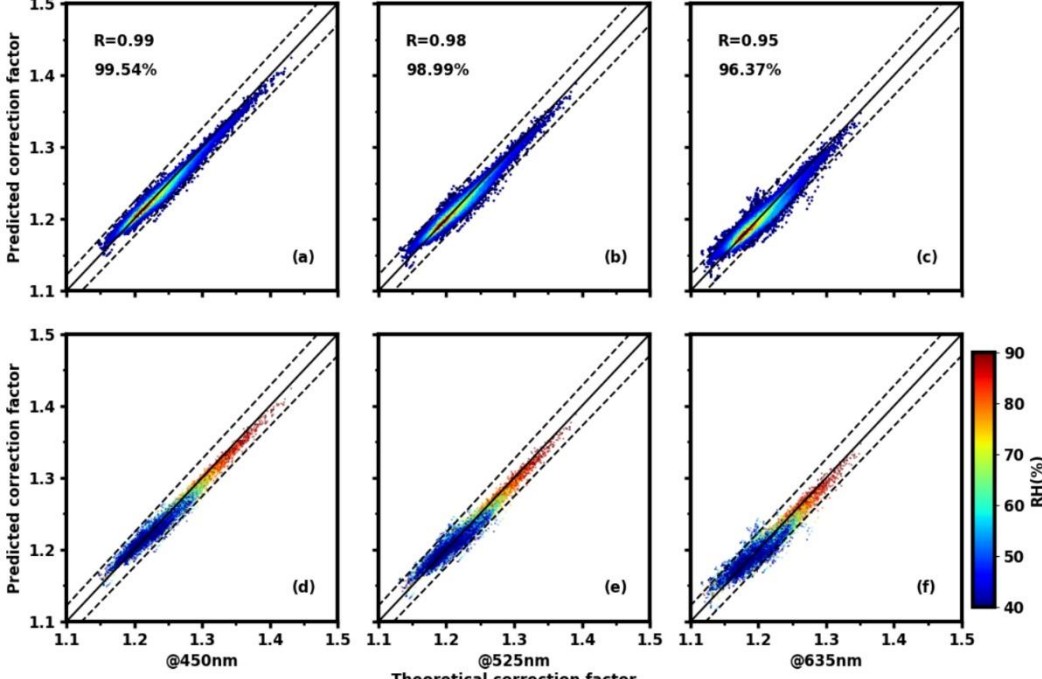

**Figure 10.** On different relative humidity conditions, the comparison of the correction factors calculated by our method and theoretical Mie-calculation values at the wavelengths of 450 nm (a), 525 nm (b), and 635 nm (c), respectively, with a black solid 1:1 line and two dashed lines representing a deviation of 2 %. The color of the data point represents data density; the warmer the hue, the denser the data point. R is the correlation coefficient, and the percentage indicates the percentile of points falling within the error range of 2 %. The data in (d), (e), (f) are the same as those in (a), (b), (c), but the color here stands for different relative humidity conditions rather than density.

## 4 Conclusions

The aerosol scattering coefficient is a significant parameter for estimating aerosol direct radiative forcing, which can be measured by nephelometers. However, nephelometers have the problem of non-ideal Lambertian light source and angle truncation, hence the observed scattering coefficient data need to be corrected. The scattering correction factor (CF) relating to the aerosol size and chemical composition is thus put forward. The most direct calibration method is to use the particle number size distribution and Mie scattering model to correct the scattering values at each wavelength under the actual nephelometer light source and the ideal light source condition. However, this method requires the auxiliary particle number size distribution and black carbon observation data, which are expensive and difficult to acquire. Later, scattering Ångström index (SAE) measured by nephelometer itself is utilized to represent aerosol particle size information, and the relationship between SAE and CF is then established. After verification, it is found that the method lacks precision and accuracy, because SAE is affected by both particle size and refractive index, while the correction factor is scarcely impacted by the refractive





index. If relative humidity increases, the particle size and refractive index may also increase accompanied by the change of aerosol water content. That is to say, the relationship between SAE and CF is complicated. Moreover, the absorption properties of sampled particles also alter the wavelength dependence of scattering, contributing to errors of this correction method for absorbing particles. Therefore, a single parameter SAE cannot predict CF very well, and our paper has proposed the new
method of nephelometer self-correction.

Under dry conditions, after analysis, SAE and HBF can represent different ranges of aerosol particle size information. With the use of the existing observation results of PNSD, black carbon and $R_{ext}$ to obtain SAE and HBF, the paper applies random forest machine learning model to figure out the relationship between CF and calculated SAE and HBF, deriving the RF predictor. With the dataset of Gucheng, the verification results show that this method is relatively accurate. The commonly
used integrating nephelometer can derive in situ scattering and backscattering coefficients at three wavelengths to calculate three SAE and three HBF. Therefore, with the use of derived RF predictor and nephelometer calculation of SAE and HBF, CF could be predicted by the nephelometer itself.

Under other relative humidity conditions, in addition to the dry aerosol particle size information, we should also consider the change in particle size and water content brought by the growth of aerosol taking up water. This paper finds that CF
increases with the increment of relative humidity and aerosol hygroscopicity. Therefore, on the basis of κ-Köhler theory, the existing observation results of PNSD, black carbon, $R_{ext}$, aerosol hygroscopicity parameter κ, and relative humidity are used to run the Mie model, obtaining the theoretical CF and 13 quantities relating to the change of CF under different RH conditions. Similarly, the random forest machine learning model is adopted to figure out the relationship between CF and the 13 quantities, deriving the RF predictor. With the dataset of Gucheng, the verification results show that the accuracy of CF obtained by this
method is very high. The humidified nephelometer system can observe scattering and hemispheric backscattering coefficients at three wavelengths under both dry and elevated RH conditions, obtaining corresponding f(RH) and $f_b(RH)$ under the nephelometer light source condition. As a result, all the 13 quantities, including six physical quantities of SAE and HBF representing dry aerosol size at each wavelength, six fitting parameters $\kappa_{sca}$ and $\kappa_{bsca}$ representing particle size-resolved hygroscopicity at each wavelength, and the relative humidity, can be directly obtained from nephelometers. Therefore, with
the use of derived RF predictor and the above 13 quantities, CF could be predicted in situ by nephelometer itself.

The strengths of our new method are summed up as follows: Under either dry or any other relative humidity conditions, the prediction performance of CF at three wavelengths is excellent. Furthermore, at each relative humidity, the accuracy of CF estimation is almost the same. All inputs can be obtained through the nephelometer's observation, achieving self-correction; that is, on the basis of ensuring accuracy of correction, there is no need for other aerosol microphysical observations.
When it comes to the weaknesses, in this study, the new method is put forward based on the datasets of continental aerosols. Moreover, due to limitations of the Mie theory, our method cannot be applied to analyse datasets which include desert and marine aerosols and hence further studies are needed. There might be errors in applying limited RF predictors to predict CF all over the world. Therefore, more field observation datasets are needed to verify and perfect this method, hopefully establishing a database of RF predictor in the future.




*Data availability.* The data used in this study is available when requesting the authors

*Author contributions.* Jie Qiu, Wangshu Tan, Gang Zhao, Yingli Yu and Chunsheng Zhao discussed the results; Wangshu Tan offered his help in the coding; Jie Qiu wrote the manuscript.


*Competing interests.* The authors declare that they have no conflict of interest.

*Acknowledgments.* This work is supported by the National Natural Science Foundation of China (41590872).

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
