# Peer review of "New correction method of scattering coefficient measurements of a three-wavelength nephelometer"

_Atmospheric Measurement Techniques, 2020_

## Author Comment (AC2)

Response to Anonymous Referee #1

*Using in situ measurements of aerosol size distribution, black carbon etc in a few stations in China, the authors developed a nice method to correct scattering coefficient measurement. This is a very interesting research and the results sound solid, so I suggest to accept this submission after a few minor revisions.*

responses: Thank you very much for your review of our manuscript. Your positive comments were very helpful and inspiring. Below we will respond to your comments one by one. Your comments are in bold italics, and my responses are in plain text. All the changes have been included in the newest version of our manuscript.

1. *L54-55, I'm a little confused why the absorption properties of particles can later the wavelength dependence of scattering*

Response: The absorption properties of particles exert impacts on the refractive indices, and the changes in the refractive index can result in a change of wavelength dependence of scattering (scattering Ångström exponent (SAE)) (Müller et al., 2011). As shown in Fig.1 of Bond et al. (2009), if particles have strong and wavelength-dependent absorption, the SAE value can be different. Therefore, the absorption properties of particles can alter the wavelength dependence of scattering (SAE) by affecting the refractive index.

2. *L81-82, I'm not confortable for this statement, CF is physically related to refractive index and particle size, SAE cannot resolve all these influences, forturnately, HBF, the simultaneous measurement with SAE, can, to some extent, provide extra information on particle size. That's it, the sentence looks like HBF is physically related to CF, but it is not, according to my understanding.*

Response: Thanks for your comment. The sentence has been rephrased in the new manuscript.

3. *The authors used a RF method, maybe it is necessary to talk about more why RF is much better than the ordinary regression method.*

Response: Thanks for your suggestion. We give an introduction to the commonly used

linear regression method and point out that this method has the main disadvantage of inaccuracy. Moreover, compared with the limitations of ordinary regression method, the advantages of RF method are stated. With the use of Gucheng data, the comparison part with simple linear regression method shown in Müller et al., (2011) has been added in Sect.3.1. The results indicate that our RF method performs better than the ordinary regression method.

Reference:

Bond, T. C., Covert, D. S., and Müller, T.: Truncation and angular-scattering corrections for absorbing aerosol in the TSI 3563 nephelometer, Aerosol Sci. Tech., 43, 866-871, doi:10.1080/02786820902998373, 2009.

Müller, T., Laborde, M., Kassell, G., and Wiedensohler, A.: Design and performance of a three-wavelength LED-based total scatter and backscatter integrating nephelometer, Atmos. Meas. Tech., 4, 1291-1303, 2011.

---

## Author Comment (AC3)

**General Comments:**

The authors describe a novel method to correct measurement errors which are inherent to the use of current commercial integrating nephelometers. This method based on machine learning sounds attractive since it is said not to "need additional observation data ". Additional data are actually needed during the machine learning phase though. The scope of applicability of the relationship between the correction factor (CF) and the two selected variables derived from the nephelometer data (the scattering Angstrom exponent, SAE, and the hemispheric backscatter fraction, HBF) is a key question that is not addressed in the manuscript. In particular, HBF is said to depend on externally mixed fraction of black carbon (Rext), while it depends also on the particle number size distribution (PNSD). Can it be demonstrated that the rule learnt by the machine to determine CF will apply at a location and/or time where different Rext and PNSD combinations lead to the same HBF for the ensemble of the aerosol? Can it be said anything about the applicability of the CF determination method described in the manuscript to nephelometer measurements performed at much less polluted locations? Other missing elements as well as the overall organisation of the manuscript make it generally quite obscure, as described below.

Response: Thank you very much for your review of our manuscript. Your comments were very helpful and constructive for improving the work's coherence and logic. It is true that we have used the particle number size distribution and BC data during the machine learning phase, but the trained random forest models are saved for researchers' further use. That is to say, when applying this method, others could use the models to directly correct the nephelometer without additional in-situ measurements. The results of Fig.3 show that there can be possible relationships between CF and the two selected parameters (SAE and HBF). Since the three parameters all relate to particle size, the machine learning method is used to derive their relationships. HBF is calculated by the ratio of backscattering to total scattering, and then Rext is derived from the inversion of HBF: Only when the difference between nephelometer calculated HBF and the Rext-assumed HBF is minimal can we obtain the Rext value (Ma et al., 2012). Therefore, it is not accurate to state that

different Rext and PNSD combinations lead to the same HBF. Even if this is the case, our machine learning method still works. The field sites we choose are all located in the Northern China Plain (NCP) where pollution is often occurred. The new Fig.2 shows that our field data can represent the background aerosol properties in NCP. Below we will respond to your comments one by one. Your comments are in bold italics, and my responses are in plain text. All the changes have been included in the newest version of our manuscript.

**Specific Comments:**

1. The method used to determine CF cannot be understood before reading steps (1) to (8) in lines 138 – 146. It would probably be useful to have an outline of Section 2.2 at the beginning of this section.

Response: Thanks very much for your comment. We have added the brief introduction of our new method in the very beginning of Sect.2.2.

2. Figure 2 shows intermediate results (indicating that satisfactory CF values cannot be obtained based on the SAE only), but there is no figure showing CF values eventually determined by the novel method vs CF values calculated using the Mie theory in dry conditions (a vast majority of nephelometers are operated in in dry conditions, in accordance with WMO-GAW recommendations).

Response: We put the verification result in the Sect.3.2. Under the dry condition, our new method of predicting CF performs well (Fig.11 in the new manuscript).

**3. Simple processes are described in details (e.g. particle hygroscopic growth) while unobvious logical steps are not precisely explained (see examples in Technical Comments).**

Response: Thanks for your comment. We have revised our manuscript accordingly.

**4. How much is the CF assessment learning depending on the assumptions about the aerosol mixing state, i.e. the fraction of purely scattering, purely absorbing, and mixed particles, including fully internally mixed aerosol?**

Response: The parameters used for machine learning are derived from Mie calculation. It is demonstrated in Ma et al. (2015, Chinese journal) that both mixing states and the change of refractive index exert little impact on the Mie calculation

results of scattering correction factor. As shown in the Table. S1 (translated from Ma et al., 2015), with the increase or decrease of refractive index, there are small relative differences of correction factors between the control group and experimental group, and it is the same case for change of mixing states. That is to say, although we set the refractive index of BC and non-absorbing as the constant value in our study, the results are still credible and accurate with small errors. Moreover, CF assessment learning barely depends on assumptions about the aerosol mixing state, too.

Table S1: Relative differences between  $CF_{550}$  calculated under different parameter assumptions and the reference value (translated from Ma et al., 2015).

| Group        | m r,NBC | m i,NBC | m r,BC | m i,BC | Mixing states     | CF 550 | Relative       |
|--------------|--------------------|--------------------|-------------------|-------------------|-------------------|-------------------|----------------|
|              |                    |                    |                   |                   |                   |                   | $CF_{550}$ (%) |
| Control      | 1.53               | 10-7               | 1.75              | 0.55              | Internal+External | 1,1174            | 0              |
| Experiment 1 | 1.50               | 10-7               | 1.75              | 0.55              | Internal+External | 1.1208            | 0.30           |
| Experiment 2 | 1.55               | 10-7               | 1.75              | 0.55              | Internal+External | 1.1150            | -0.21          |
| Experiment 3 | 1.53               | 10-7               | 1.50              | 0.55              | Internal+External | 1.1190            | 0.14           |
| Experiment 4 | 1.53               | 10-7               | 2.00              | 0.55              | Internal+External | 1.1157            | -0.15          |
| Experiment 5 | 1.53               | 10-7               | 1.75              | 0.44              | Internal+External | 1.1169            | -0.04          |
| Experiment 6 | 1.53               | 10-7               | 1.75              | 0.66              | Internal+External | 1.1177            | 0.03           |
| Experiment 7 | 1.53               | 10-7               | 1.75              | 0.55              | Internal          | 1.1178            | 0.04           |
| Experiment 8 | 1.53               | 10-7               | 1.75              | 0.55              | External          | 1.1171            | -0.03          |

5. It is not stated if the CF determination method described would apply to TSI 3563 instruments, nor how measurements performed with a TSI 3563 (at least in campaign #5) were used in the machine learning process regarding the Aurora 3000 instrument.

Response: Our new method works for three-wavelength nephelometers, and hence it can be applicable to both Aurora 3000 and TSI 3563. This paper takes the Aurora 3000 as an example to introduce this method, and researchers can follow the method to figure out how TSI 3563 measurements perform. As for the TSI 3563, the truncation angle and the angular intensity distribution are different from those of Aurora 3000. That is to say, the machine learning process should be redone with the parameters calculated under the condition of TSI 3563 nephelometer's light source. It is not reasonable to use the random forest model trained for Aurora 3000 to predict the correction factor for TSI 3563. In our group, there is something wrong with TSI 3563 and Aurora 3000 is the only nephelometer that we use for now. Therefore, this

paper takes Aurora 3000 rather than TSI 3563 as an example.

**Technical Comments:**

**1. Line 23: "... the aerosol direct radiative forcing varies ..." across what, as a function of what? Is this a range of uncertainty or variability?**

Response: Sorry for the ambiguity. This is a range of uncertainty and we have added "the uncertainty of" in the new manuscript.

2. Line 25: Aerosol direct radiative forcing also depends on the aerosol HBF and vertical profile (or at least the integrated aerosol optical depth). The 4 variables (aerosol single scattering albedo, extinction coefficient, aerosol scattering coefficient, and absorbing coefficient) are equivalent. Knowing 2 of them is enough. Since this manuscript is about integrating nephelometers (which measure scattering), I would suggest to stick to "scattering and absorbing coefficients" Response: Thanks for pointing this out. We have revised this part.

**3. Line 50: suggestion: "Bond et al. (2009) found that SAE is also affected by the particle refractive Index."**

Response: Thanks for your suggestion. Revised as suggested.

**4. Line 50 – 56: please consider streamlining: the sentences referring to Bond et al (2009) are redundant.**

Response: We have revised this part in the new manuscript.

**5. Line 70: suggestion: "our number size distribution measurements cover a wide range of 10-1000 nm, ...."**

Response: Thanks for your suggestion. Change made.

**6. Line 71: Table 1: which nephelometers were used in campaigns 1-4?**

Response: We did not own nephelometers at that time, and hence no nephelometers were used in campaigns 1-4.

7. Line 88: "...three types of particles: ". The composition (i.e. chemical composition) controls the refractive index. What was the refractive index selected

**for the absorbing material?**

Response: We selected black carbon as the absorbing material and the refractive index is set to be 1.80-0.54i (Ma et al., 2012). Information has been added in the new manuscript.

8. Lines 97-100: this section is confusing. The variations in the SAE as a function of the particle diameter directly result from the Mie theory, and does not support the last sentence starting with "Therefore" (line 99): on which basis are particles in the size range 100-200 nm stated not to contribute to the overall SAE values? Is it meant that they do not contribute much to SAE variations?

Response: Yes, it is certain that SAE varies with particle diameter according to the Mie theory. We aim to demonstrate that, for particles in the size range 100-300 nm, the SAE variations are rather small. For instance, 100 nm particle can have the similar SAE value as 150 nm particle. If we use SAE values in this size range of 100-300 nm, particle diameters cannot be well represented. Sorry for this unclear statement. We have revised this part in the new manuscript.

9. Line 107-109: "aerosol particles show a noticeable feature of HBF decreasing with the increment of particle size. However, when the particle becomes larger than 300 nm, the HBF is almost unchanged." is again a direct consequence of the Mie scattering theory. And again the logical connection with the last sentence "HBF can represent the size information of particles smaller than 300 nm" is unclear. It was probably meant that HBF variability is mostly sensitive to the concentration of particles in the 100-300 nm size range.

Response: Yes, that is what we try to demonstrate. We have revised this part to avoid misunderstanding.

**10. Line 123: rather HBF is sensitive to Rext**

Response: Thanks for pointing this out. Change made.

**11. Line 149: RF is not defined.**

Response: Thanks for your comment. We have defined it in the new manuscript.

**12. Figure 5: the diagram omits to mention that Mie calculations are performed**

**using both the actual nephelometer light source characteristics and an ideal light source, which is essential for determining CF.**

Response: Both diagrams are updated based on your suggestion.

**13. Line 167-168: RH instead of RH' in the denominator**

Response: Sorry for the ambiguity. It should be a comma. We have deleted it in the revised manuscript.

**14. Line 171: "f (RH) and fb(RH) values "**

Response: Thanks for your comment. Revised as suggested.

**15. Line 175: C(RH) is not precisely defined. It can be guessed afterwards that C = CF.**

Response: Thanks for pointing this out. C means CF actually and C(RH) is the hygroscopic growth of CF. We have rephrased this sentence in the new manuscript.

**16. Line 231: information is missing to support the statement "which improves the accuracy of the CF estimation in the dry state": "improves" compared to what?**

Response: We stated at the beginning of the sentence "compared with the traditional correction method".

**17. Line 249-250: this is quite obvious since increasing RH and increasing hygroscopicity have the same effect on the particle sizes (increased diameters).**

Response: Yes, this sentence is obvious to some extent, but we need one conclusion at the end of the paragraph. We have rephrased the sentence according to your comment.

**18. Line 271: suggestion: "essential" rather than "significant"Response: Thanks for your suggestion. Revised as suggested.**

19. Line 273: The sentence "The scattering correction factor (CF) relating to the aerosol size and chemical composition is thus put forward" is unclear. Is it meant: 'The correction factors (CF) to be applied depend on the aerosol particle number size distribution and chemical composition."?

Response: Sorry for the unclear statement. What we want to stress is that the

scattering correction factor (CF) is thus put forward. That CF relates to the aerosol size and chemical composition is the additional information used as postpositive attributes. This sentence is rephrased as "The scattering correction factor (CF) is thus put forward and it depends on the aerosol size and chemical composition."

20. Line 274-285: The description of the Mie calculation method is obscure. However, a clear and concise description is needed, since it is the basis for the machine learning process.

Response: We have deleted some sentences to make this part more concise.

**21. Line 286: Suggestion: "SAE and HBF provide information on the aerosol particle size distribution for different size ranges (…)". The size ranges should be specified in brackets.**

Response: Thanks for your suggestion! We have added "(300-1000 nm for SAE and 100-300 nm for HBF)" in the revised manuscript.

**22. Line 293: The first sentence should state that this paragraph regards 'humidified nephelometer measurements".**

Response: We have added "the humidified nephelometer system is utilized" in the revised manuscript.

Reference:

- Ma, N., Zhao, C. S., Müller, T., Cheng, Y. F., Liu, P. F., Deng, Z. Z., Xu, W. Y., Ran, L., Nekat, B., van Pinxteren, D., Gnauk, T., Müller, K., Herrmann, H., Yan, P., Zhou, X. J., and Wiedensohler, A.: A new method to determine the mixing state of light absorbing carbonaceous using the measured aerosol optical properties and number size distributions, Atmos. Chem. Phys., 12, 2381-2397, doi:10.5194/acp-12-2381-2012, 2012.
- Ma, N., Zhou, X. J., Yan P. and Zhao, C. S.: A method of correcting the measurements of TSI 3563 nephelometer, Journal of Applied Meteorological Science (Chinese journal), 26(01), 15, doi: 10.11898/1001-7313.20150102, 2015.

---

## Author Comment (AC4)

**Response to Anonymous Referee #2**

**General comments:**

The authors introduce a new correction method to correct for the truncation error of the Aurora3000 nephelometer. This method uses the Angstrom exponent and also the hemispheric backscattering coefficient and is based on training a random forest machine learning model. To the reviewer's knowledge, the method is new and could be a step forward.

**However, the reviewer has major concerns about the description of the model and the presentation and interpretation of results.**

Response: Thank you very much for your review of our manuscript. We sincerely appreciate the efforts you have put in the review process and have revised our manuscript according to your comments. Below we will respond to your comments one by one. Your comments are in bold italics, and my responses are in plain text. All the changes have been included in the newest version of our manuscript.

The role of the field measurements in this manuscript is not clear. In the absence of a complete, albeit simple, classification of the field data using SAE and SSA, it is questionable whether the data provide a sufficient basis for initialising the model. For example, it is not clear how strong the light absorption of the simulated aerosols are. Any information on single scattering albedo or the imaginary part of the refractive index are missing.

Response: All the field campaign sites are located in the Northern China Plain and the field data can represent the background aerosol properties there. As shown in Fig.S1, the single scattering albedo (SSA) of eight datasets varies between 0.235 and 0.997, and the scattering Ångström exponent (SAE) also covers a wide range, indicating that the field measurements used in this manuscript is sufficient to initialize the model.

Fig.S1: The probability density distribution of SSA and SAE.

Furthermore, the reviewer finds it difficult to distinguish between measurement (PNSD) and speculative assumptions (refractive indices or kappa) in both models (dry and different RH conditions). For simplicity, the simulation study could also have been carried out with synthetic data for clearly defined aerosol types. The description of the model calculations are often imprecise, as important parameters such as refractive indices used from etc. are not specified.

Response: Both the measurement and speculative assumptions are needed in order to derive parameters to train the model. The reason why we use the in-situ measurements rather than synthetic datasets is that we want to train the machine learning model for the in-situ nephelometer correction. With these in-situ datasets, we can obtain a model that can be better representative of in-situ measurements, hence we can obtain better correction results. We assumed that the aerosols were composed of absorbing black carbon and non-absorbing materials, and their refractive index is set to be 1.80-0.54i (Ma et al., 2012) and  $1.53-10^{-7}$ i (Wex et al., 2002), respectively. Refractive indices and single scattering albedo are addressed in the new manuscript.

Chapter 3 points out the performance of the new model. Why are results of the new model only shown for data from the Gucheng measurement campaign? Why were experimental data not used to show the performance of the new algorithm for dry conditions by a closure experiment of the light scattering coefficient? And more

important, why haven t the authors shown how their approach compares to the simple linear parameterization shown in Mueller et al., (2011)?

Response: When training the machine learning model, we need to choose the training datasets first. The more training datasets we use, the better the trained model can perform. Therefore, we split eight datasets into seven training datasets and one test dataset (Gucheng), and we can only use the test dataset to verify our new model. A closure experiment of the light scattering coefficient has been done and for more details please refer to Fig.S1 in the supplement of Yu et al. (2018). Our revised manuscript has added the comparison with the simple linear parameterization shown in Müller et al. (2011), and it can be seen that this linear regression method is less accurate than our method.

**Specific comments:**

**1. Line 36: What parameter is mentioned?**

Response: It is the "scattering correction factor" in the sentence.

**2. Line 39: What methods have been proposed in Mueller et al (2011)?**

Response: In Müller et al. (2011), the two methods, also mentioned in the introduction of our manuscript, are correction using measured size distributions and correction using scattering Angström exponents. Müller et al. (2011) did not put forward a new correction method and what they focused on is to provide parameterizations for angular sensitivity functions of the Aurora 3000.

3. Line 56: Figure 5 in Mueller et al (2011) suggests that a simple linear function is not sufficient. Unfortunately, this was not discussed further in Mueller et al. (2011). Response: Yes. Müller et al. (2011) followed the simple linear regression method put forward by Anderson and Ogren (1998). They only utilized one parameter (SAE) to do the regression analysis. In the Sect.3.1 of our revised manuscript, using Gucheng data, we find out that this linear method lacks accuracy. Inspired by their work, the paper uses more relating parameters to predict CF and the verification results show better performance.

4. Chapter 1: In general, the description of the state of the knowledge is little vague. How large are uncertainties when using the simple parameterizations of Anderson

**(1998) and Mueller (2011)?**

Response: Anderson and Ogren (1998) and Müller (2011) used the same method that is correction using scattering Angström exponents. As for different datasets, the uncertainties could be different. We have used Gucheng data to compare this linear regression method with ours and added this part to Sect. 3.1 in our revised manuscript.

5. Line 70 and Figure 1: Just taking a large set of total number concentrations as an argument that a large number of possible aerosol types have been covered is not sufficient. Furthermore, no evidence of a coarse mode particle can be seen in the particle size distributions. The large range of scattering Angström exponents (see Figure 2) suggest that could be are cases with a significant coarse mode volume fraction.

Response: As seen in Fig.S2 (also added as Fig.2 in the revised manuscript), SSA of eight datasets varies a lot and the measurements can represent the background aerosol properties in the Northern China Plain. Our datasets barely include the coarse mode particles taking the location and date of campaign into account. Due to the limitations of Mie model, our method is not suitable for coarse mode particles (discussed at the end of the paper), too. We have deleted "including most continental aerosol types" to avoid misunderstanding.

Fig.S2: The SSA of eight datasets.

6. Line 89: A core radius of 35 nm might be too small to represent internally mixed aged particles. Furthermore, a constant core size also means that the volume fraction of absorbing material and the single scattering albedo decreases with increasing particle size. What does this mean for the interpretation of Figure 3? What range of single scattering albedos is covered with this model?

Response: As shown in Fig.S1 and Fig.S2, the single scattering albedo (SSA) of eight datasets varies between 0.235 and 0.997. This paper made the assumption of three independent cases including all scattering particles, all absorbing particles, and all core-shell mixed particles of which core diameter is 70 nm. Ma et al. (2012) pointed out that the freshly emitted LAC particles are assumed to be distributed with geometric average diameter of 50 nm. Therefore, diameter of 70 nm can represent the internally mixed aged aerosols to some extent. Here we also compare the results between core diameter of 70 nm and that of 100 nm (Fig.S3) and their general trend is similar.